# Don't let your Discriminator be fooled

**Brady Zhou**
Department of Computer Science
University of Texas
`brady.zhou@utexas.edu`

**Philipp Krähenbühl**
Department of Computer Science
University of Texas
`philkr@cs.utexas.edu`

## Abstract

Generative Adversarial Networks are one of the leading tools in generative modeling, image editing and content creation. However, they are hard to train as they require a delicate balancing act between two deep networks fighting a never ending duel. Some of the most promising adversarial models today minimize a Wasserstein objective. It is smoother and more stable to optimize. In this paper, we show that the Wasserstein distance is just one out of a large family of objective functions that yield these properties. By making the discriminator of a GAN robust to adversarial attacks we can turn any GAN objective into a smooth and stable loss. We experimentally show that any GAN objective, including Wasserstein GANs, benefit from adversarial robustness both quantitatively and qualitatively. The training additionally becomes more robust to suboptimal choices of hyperparameters, model architectures, or objective functions.

## 1 Introduction

Generative adversarial networks (GANs) (Goodfellow et al., 2014a) are at the forefront of generative modeling. They cast generative modeling as a never ending duel between two networks: a generator network produces synthetic samples from random noise and a discriminator network distinguishes synthetic from real data samples. GANs produce visually appealing samples, but are often hard to train. Much recent work, tries to stabilize GAN training through novel architecture (Radford et al., 2015; Denton et al., 2015) or training objectives(Mao et al., 2017; Arjovsky et al., 2017; Gulrajani et al., 2017).

In this paper, we show that GAN objectives significantly stabilize, if the discriminator is robust to adversarial attacks. Specifically, we show that a robust discriminator leads to a robust minimax objective for the generator irrespective of the training objective used. In addition, any training objective is smooth and Lipschitz as long as the discriminator is Lipschitz. Finally, we show that this robustness does not need to be enforced for every single input of the discriminator, but rather just in expectation over the generated samples. We present two new regularization terms, borrowed from the adversarial training literature. Both terms ensure the robustness of the discriminator. They are easy to optimize and can be added to any existing GAN objective.

Our experiments show that adversarial robustness both improves the visual quality of the results, as well as stabilizes the training procedure across a wide range of architectures, hyper-parameters and training objectives. We will publish the code and data used to perform our experiments upon acceptance.

## 2 Related Works

The scope of generative modeling changed significantly with the emergence of photo realistic image generation like Generative Adversarial Networks (GANs) (Goodfellow et al., 2014a), Variational Auto-Encoders (Kingma & Welling, 2013), or Pixel Convolutional Networks (Oord et al., 2016). The visual quality of these generative models enabled applications such as, generative image processing (Pathak et al., 2016), image editing (Zhu et al., 2016), and image translation (Isola et al., 2017; Zhu et al., 2017). They also started the race for ever prettier generated images. Two major areas of improvement are architectures (Radford et al., 2015; Denton et al., 2015) and loss functions (Mao et al., 2017;

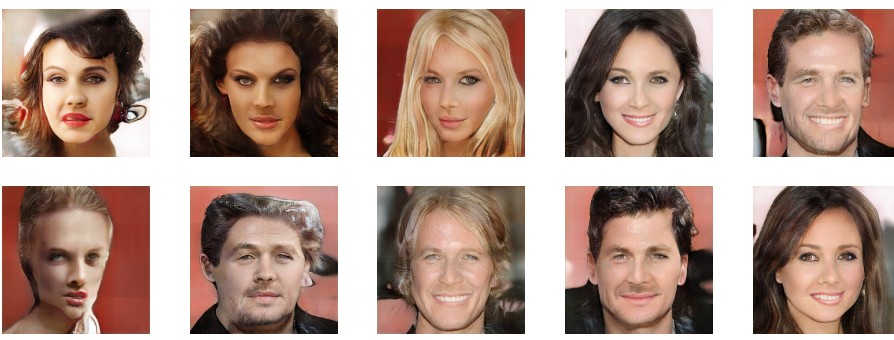

standard discriminator          robust discriminator

Figure 1: Results from four generative adversarial models. The leftmost model is trained with a standard discriminator, while the models on the right use an increasingly more robust discriminator. We show that a robust discriminator leads to a smoother and more stable training objective, resulting in a better generative model. Best viewed on screen.

Arjovsky et al., 2017; Gulrajani et al., 2017). Denton et al. (2015) showed that a Laplacian pyramid generator greatly improves the quality of the generated images, while Radford et al. (2015) showed impressive results by carefully balancing the expressive power of the generator and discriminator networks. In this work, we use Radford *et al.* (Radford et al., 2015) architecture, and solely focus on the loss function.

GANs come in three popular loss functions: the original Jensen-Shanon divergence objective (Goodfellow et al., 2014a), least squares GANs (LSGAN) (Mao et al., 2017), and Wasserstein distance (WGAN) (Arjovsky et al., 2017). Each objectives minimizes special case of a f-divergence between the generative and data distribution (Nowozin et al., 2016). Arjovsky et al. (2017) augment the GAN objective with a smoothness condition in the form of a Lipschitz-1 constraint on the discriminator. However, enforcing a Lipschitz-1 discriminator is difficult. Arjovsky et al. (2017) clip the weights of the discriminator to enforce the Lipschitz condition, while Gulrajani et al. (2017) apply a penalty on the gradient of the discriminator. The gradient penalty (WGAN-GP) generally yields better results, however it looses some of the appealing theoretical properties of WGANs. Arjovsky *et al.*show that in theory most GAN objective are non-continuous and non-Lipschitz, while the Wasserstein objective is Lipschitz, smooth, and more stable to optimize. In this paper, we extend their analysis and show that any GAN objective can be made smooth as long as the discriminator is robust to adversarial perturbations. In addition, the discriminator only needs to be robust to adversarial attacks in expectation over all generated samples. A simple penalty function is sufficient to carry the theoretical smoothness results of WGAN other GAN objectives.

Our theoretical analysis is related to Bottou et al. (2017), who study the geometry to common distance measures for generative modeling. We build on their analysis and show that any GAN objective is Lipschitz and smooth, as long as the generator and discriminator are Lipschitz.

**Adversarial Examples**   Despite success of deep networks, their differentiable nature makes them vulnerabilities to adversarial attacks. Small perturbation of the input can significantly change the output of a network (Szegedy et al., 2013). The initial work of Szegedy et al. (2013) set off an arms race between novel attack methods (Goodfellow et al., 2014b; Carlini & Wagner, 2017; Papernot et al., 2016a; Moosavi Dezfooli et al., 2016) and defenses (Kannan et al., 2018; Papernot et al., 2016b) against these attacks. The goal of an attacker is to find an perturbation in an attack region that yields a large change in output. Some of the fastest attacks simply perturb the input in the gradient direction of the network (Goodfellow et al., 2014b), while more complex attacks optimize for an attack vector (Carlini & Wagner, 2017). Many attacks can be directly used in a defense through dataset augmentation (Papernot et al., 2016b). In this work, we primarily use the fast gradient methods of Goodfellow et al. (2014b), in particular we use the normalized gradient attack.

## 3 PRELIMINARIES

Let $z \sim P_Z$ be a sample from a noise distribution $P_Z$, e.g. uniform noise. Let $x \sim P_R$ be a sample form a data distribution $P_R$. The goal of sampling based generative modeling is to find a function $G(z)$ that maps samples from $P_Z$ to $P_R$. More specifically, we want $P_Y = \{y = G(z) | z \sim P_Z\}$ to closely resemble $P_R$.

**Generative Adversarial Networks** A generative adversarial network (GAN) optimizes a two player game between a generative model $G$ and a discriminator $D$. The generator $G$ maps noise to data samples. A discriminator $D$ then judges if the transformed noise is close enough to the true data distribution $P_R$. A GAN jointly optimizes both generator and discriminator in a minimax game. The GAN objective $L$ minimizes a generator $G$ and maximizes a discriminator $D$ over

$$L(G, D) = \mathbb{E}_{x \sim P_R}[f(-D(x))] + \mathbb{E}_{z \sim P_Z}[f(D(G(z)))], \tag{1}$$

where $f$ is differentiable, usually concave, loss function. The generator minimizes

$$\ell(G) = \max_{D \in \mathcal{D}} L(G, D). \tag{2}$$

In practice, both generator $G(z; \theta)$ and discriminator $D(x; \phi)$ are deep neural networks with parameters $\theta$ and $\phi$ respectively. The discriminator $D$ produces a single scalar output, which is then passed through the loss function $f$. Different choices of $f$ lead to different GAN models. The original GAN objective uses a sigmoid log likelihood loss for $f(x) = -\log(1 + \exp(-x))$, and corresponds to the Jensen-Shannon divergence between data and generator (Goodfellow et al., 2014a). A Euclidean loss $f(x) = -\frac{1}{2}(1 + x)^2$ leads to least squares GAN (LSGAN) (Mao et al., 2017). A Lipschitz-1 discriminator $D$ and identity loss functions $f(x) = x$ corresponds to the Wasserstein distance in WGAN (Arjovsky et al., 2017). In all these examples f is concave, and for GAN and LSGAN non-positive.

**Adversarial attacks** We follow the standard adversarial attack definition in literature (Carlini & Wagner, 2017; Kannan et al., 2018).

**Definition 3.1** (robustness). A function $h(x)$ is robust to adversarial perturbations $\Delta$ for an input $x$ if and only if

$$|h(x) - h(x + \Delta)| < \varepsilon \qquad \text{for all } \|\Delta\|_p < \delta,$$

where the p-norm $\| \cdot \|_p$ defines the local attack region.

For simplicity of the analysis, we focus on distance norms $| \cdot |$ instead of general distance functions of Carlini & Wagner (2017). Our definition includes logit pairing of Kannan et al. (2018).

Definition 3.1 focuses on individual inputs $x$ drawn from an empirical data distribution of training or testing images. We extend this definition to generative distributions $G(z)$.

**Definition 3.2** (expected robustness). A function $h$, e.g. a discriminator, is robust to adversarial perturbations for a generative distribution $G(z)$ if and only if

$$E_{z \sim P_Z}[|h(G(z)) - h(G(z) + u(z))|] < \varepsilon \qquad \text{for all functions } u \text{ with} \|u(z)\|_p < \delta,$$

where the p-norm $\| \cdot \|_p$ defines the local attack region, and $d$ measures the distance between the network outputs.

This definition only requires robustness in expectation, but not for every single sample $z \sim P_Z$. A discriminator $D$ is robust if it satisfies definition 3.2 for $h = D$. A discriminator loss $f$ is robust for $h = f \circ D$. For notational simplicity, we define the additive combination of a generator $G$ and a perturbation function $u$ as $G + u$ where $(G + u)(z) = G(z) + u(z)$.

## 4 ROBUST GENERATIVE ADVERSARIAL NETWORKS

In this section, we show that the GAN objective is robust to adversarial perturbations, as long as the discriminator is robust. This property even holds as the discriminator adapts with the perturbation applied. We first analyze the case when the loss of the discriminator is robust to an adversarial attack, and then study the robustness of the discriminator directly.

**Theorem 4.1** (robust loss). If the loss of a discriminator $D$ is robust to adversarial perturbations $\|u(z)\|_p < \delta$ in expectation

$$E_{z \sim P_Z}\left[|f(D(G(z))) - f(D(G(z) + u(z)))|\right] < \varepsilon$$

then the adversarial objective $\ell$ is robust

$$|\ell(G) - \ell(G + u)| < \varepsilon$$

*Proof.* Let $D_G$ be the optimal discriminator for generator $G$. For this discriminator we have

$$L(G, D_G) = \max_D L(G, D) \geq L(G, D_{G+u}), \tag{3}$$

by definition. This allows us to bound the difference in the objective for a generator $G$ and its perturbation $G + u$:

$$\begin{aligned}
\ell(G) - \ell(G + u) &\leq L(G, D_G) - L(G + u, D_G) && \text{Inequality 3} \\
&= \mathbb{E}_{z \sim \mu_z}[f(D_G(G(z))) - f(D_G(G(z) + u(z)))] < \varepsilon && \text{linearity of expectation}
\end{aligned}$$

We can equivalently bound $\ell(G + u) - \ell(G) < \varepsilon$, and obtain $|\ell(G) - \ell(G + u)| < \varepsilon$. $\qquad\square$

The above proof uses the (global) optimality of the discriminator in Equation (3). However, the bound holds for any discriminator that improves the objective, without reaching a local or global minima, and could thus be generalized.

Next, we focus our attention to robust discriminators. Here, we rely on an additional assumption that all loss functions $f$ we consider are concave. This is the case for the three most popular architectures: original GAN, LSGAN and WGAN.

**Theorem 4.2** (robust discriminator). For a concave loss $f$ and a discriminator $D$ that is robust to perturbations $\|u(z)\|_p < \delta$ in expectation

$$E_{z \sim P_Z}\left[|D(G(z)) - D(G(z) + u(z))|\right] < \varepsilon$$

the adversarial objective $\ell$ is robust

$$|\ell(G) - \ell(G + u)| < \varepsilon C$$

where $C = E_{z \in P_Z}\left[\max(|f'(D(G(z)))|, |f'(D(G(z) + u(z)))|)\right]$ is the expected loss gradient.

*Proof.* The proof follows directly from the definition of concavity $f(a) - f(b) \leq f'(a)(a - b)$ and the fairly conservative bound $|f(a) - f(b)| \leq \max(|f'(a)|, |f'(b)|)|a - b|$. In expectation, this bound reduces the robust discriminator to a robust loss

$$E_{z \sim P_Z}\left[|f(D(G(z))) - f(D(G(z) + u(z)))|\right] \leq E_{z \sim P_Z}\left[|D(G(z)) - D(G(z) + u(z))|\right] C < \varepsilon C.$$

The rest of the proof follows Theorem 4.1. $\qquad\square$

For the original GAN $C < 1$, for WGAN $C = 1$, and for LSGAN $C \leq E_{z \in P_Z}\left[|1 + D(G(z))|\right] + \varepsilon$, where $|1 + D(G(z))|$ is directly minimized in the objective and is close to zero.

Theorem 4.1 and Theorem 4.2 have some interesting implications for general GANs. First, any GAN that is trained with a robust discriminator or loss has a robust and hence smooth objective. This directly extends the theoretical properties of WGANs (Arjovsky et al., 2017) to other GAN model. It further relaxes the strict Lipschitz constraints to adversarial robustness in expectation. This is much easier to enforce in practice as we will show in the next section.

Second, Theorem 4.2 allows us to analyze the continuity and Lipschitzness of any GAN objective. In particular, we can show any GAN objective is continuous or Lipschitz in its parameters as long as the discriminator and generator are continuous or Lipschitz. This is in direct contradiction to Arjovsky et al. (2017), that show only WGAN to be continuous. However, their counter example relies on a discontinuous discriminator with infinitely large weights. See supplemental material for a detailed derivation and proof.

Robust discriminators or losses can be hard to train in practice, as they require constrained optimization or a carefully tuned architecture. Next, we show how a regularized discriminator objective can lead to a robust discriminator without the need for any constraints.

### 4.1 TRAINING A ROBUST DISCRIMINATOR

Our goal is to train a discriminator $D$ such that it is robust to adversarial perturbations in expectation. We do this by augmenting the original discriminator training objective with an additional adversarial regularization

$$\rho(D, G) = E_{z \sim P_Z} \left[ \max_{v:\|v\|_p < \delta} (D(G(z)) - D(G(z) + v))^2 \right]. \tag{4}$$

In adversarial defense, this is known as distillation (Papernot et al., 2016b) or logit pairing (Kannan et al., 2018), where $v$ is the attack vector in an attack region $\|v\|_p < \delta$. For the main evaluation, we find the attack vector using the fast normalized gradient attack (Goodfellow et al., 2014b): $v = \frac{\nabla D(G(z))}{\|\nabla D(G(z))\|_2}$. Additional results using other attack methods are in the supplementary material.

This adversarial regularization is sufficient to ensure a robust discriminator.

**Theorem 4.3** (regularized robustness). For a non-positive loss $f$ and a regularized discriminator objective

$$\underset{D \in \mathcal{D}}{\text{maximize}}\, L(D, G) - \lambda \rho(D, G)$$

the optimal discriminator $D^*$ is robust

$$E_{z \sim P_Z} \left[ |D^*(G(z)) - D^*(G(z) + u(z))| \right] \leq \sqrt{\frac{-2f(0)}{\lambda}}$$

for all $\|u(z)\|_p < \delta$.

*Proof.* We know that the objective value of the optimal discriminator is larger than the objective of a constant zero discriminator:

$$L(D^*, G) - \lambda \rho(D^*, G) \geq L(0, G) - \lambda \rho(0, G) = L(0, G)$$
$$\lambda \rho(D^*, G) \leq L(D^*, G) - L(0, G) \leq -L(0, G) = -2f(0),$$

where the last inequality holds due to the non-positivity of the objective. Using Jensen's inequality we can further reduce

$$\max_{u:\|u(z)\|_p < \delta} E_{z \sim P_Z} \left[ |D^*(G(z)) - D^*(G(z) + u(z))| \right]^2 \leq$$

$$\max_{u:\|u(z)\|_p < \delta} E_{z \sim P_Z} \left[ |D^*(G(z)) - D^*(G(z) + u(z))|^2 \right] = \rho(D^*, G) \leq \frac{-2f(0)}{\lambda}.$$

$\square$

As we increase the weight $\lambda$ of the robustness term, the bound in Theorem 4.3 tightens and the discriminator is more robust to adversarial perturbations. A similar bound can be derived for an absolute instead of squared loss $\rho$.

While the adversarial regularization $\rho$ yields all the nice theoretical benefits of a robust discriminator, in practice it often does not provide enough regularization. A much stronger regularization is to match the features $\phi(x)$ of the penultimate layer of the discriminator network $D(x) = w^\top \phi(x)$. We call this robust feature matching (RFM). Our robust feature matching minimizes

$$\rho_r(D, G) = E_{z \sim P_z} \left[ \max_{v:\|v\|_p < \delta} \|\phi(G(z)) - \phi(G(z) + v)\|_2^2 \right] + \alpha \|w\|_2^2 \tag{5}$$

where $v$ is the attack direction, $w$ is the weight of the last linear layer, and $\alpha \in \mathbb{R}^+$ is a hyperparameter. We use $\delta = 0.05$ and $\alpha = 10^{-4}$ in all our experiments. Similar to Theorem 4.3, robust feature matching ensures the robustness of the optimal discriminator

$$E_{z \sim P_Z} \left[ |D^*(G(z)) - D^*(G(z) + u(z))| \right] \leq \frac{-2f(0)}{\sqrt{\alpha \lambda}}.$$

For a detailed proof see supplement.

Next, we will give some intuition on how adversarial defense changes the loss landscape of a GAN on a small toy example.

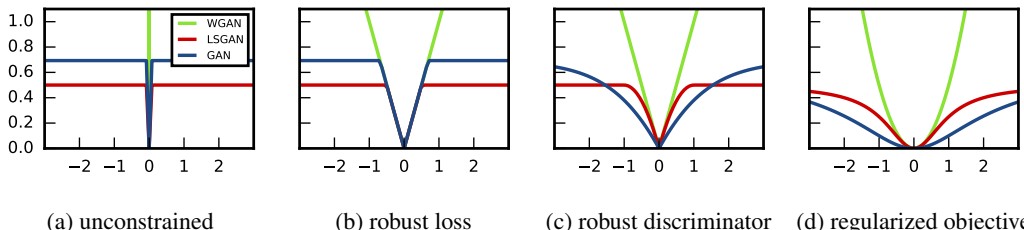

(a) unconstrained    (b) robust loss    (c) robust discriminator    (d) regularized objective

Figure 2: Dirac-GAN objective value as a function of the generator parameter $\theta$ for linear (WGAN), least squares (LSGAN) and Jensen-Shanon (GAN) objectives: (a) without constraints, (b) constrains on the loss function (Thm 4.1), (c) constrains the discriminator output (Thm 4.2), (d) a regularizer on the discriminator output (Thm 4.3). Adding adversarial robustness to the discriminator smoothes out any objective. Best viewed in color.

## 4.2 A Toy Example

We verify the above robustness properties on the simple one dimensional Dirac-GAN of Mescheder (2018). It allows us to separate practical optimization issues from the form and regularity of the loss function. We can easily optimize the discriminator of Dirac-GAN to convergence with or without robustness constraints or penalties. In the Dirac-GAN, the true data distribution $p_D$ is represented by a Dirac distribution centered at zero. The generator produces samples from another Dirac distribution centered at $\theta$, where $\theta$ is the only learnable generator parameter. The discriminator is a linear function $D(x) = \psi x$ with a single scalar parameter $\psi$. This experimental setup is analogous to the two dimensional toy example of Arjovsky et al. (2017). The GAN objective of Dirac-GAN reduces to $L(G) = \max_\psi f(\psi\theta) + f(0)$. We plot this objective in Figure 2a for unconstrained GAN, LSGAN and WGAN respectively. Note, that all objectives are zero if the data and generative distributions match and constant otherwise. Without any regularization, neither one of these objectives will provide a meaningful gradient signal.

Adding a robust loss Figure 2b, Theorem 4.1, or a robust discriminator Figure 2d, Theorem 4.2, will smooth out all loss functions and provide a meaningful gradient signal throughout training. Finally, the regularized loss shows an equally smooth loss curve. This serves as an illustrative example of the effect of robustness on the overall loss function of a GAN.

## 5 Experiments

We perform all our experiments on the MNIST (LeCun et al., 1998), CIFAR10 (Krizhevsky, 2009) and CelebA (Liu et al., 2015) datasets. We use a slightly modified DCGAN (Radford et al., 2015) architecture. All convolutional blocks are replaced with residual blocks (He et al., 2016), the generator employs batch normalization (Ioffe & Szegedy, 2015) and ReLU nonlinearities, while the discriminator uses instance normalization (Ulyanov et al., 2016) and Leaky ReLU. We train using a weight decay term $\lambda = 10^{-4}$, with batch size $n = 64$ and optimize using ADAM (Kingma & Ba, 2014) with $h = 2 \cdot 10^{-4}$ and $\beta_0 = 0$, $\beta_1 = 0.9$ for 50 epochs on CIFAR10 and 25 epochs on CelebA. We use a latent vector of dimension $z = 128$ and use a unit Gaussian for our sampling distribution. For complete training and architecture details see supplement.

**Robust GAN training** We compare original Jenson-Shannon divergence loss (GAN) (Goodfellow et al., 2014a), the least squares loss (LSGAN) (Mao et al., 2017), and the Wasserstein distance (WGAN) (Arjovsky et al., 2017) on CIFAR10 and CelebA. Along with these losses, we add a regularization method: Instance Noise (IN) (Sønderby et al., 2016), Gradient Penalty (GP) (Gulrajani et al., 2017), Adversarial Regularization (AR) $\rho$, and Robust Feature Matching (RFM) $\rho_r$. We tuned the hyper-parameters for each combination of loss functions and robustness terms (or none).

We present quantitative results in terms of Fréchet Inception Distance (FID) (Heusel et al., 2017), as well as qualitative results. The FID score measures the distance between two Gaussian distributions that are estimated from the deep feature statistics from the real and generated distributions. Distributions that are close have a lower FID score, distributions that are large have a higher score. Heusel

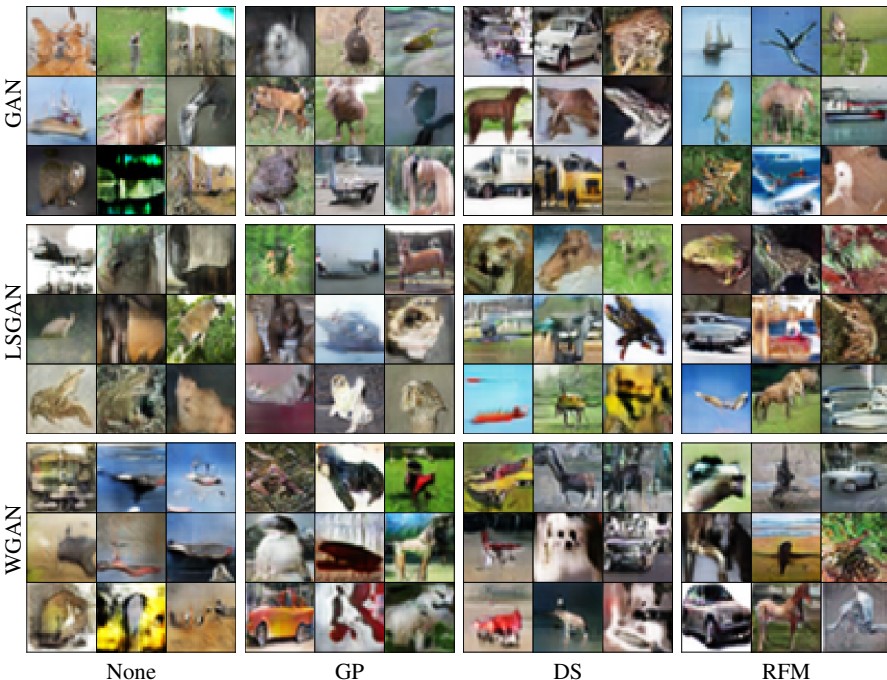

Figure 3: Random image samples from CIFAR10 for various GAN objectives (GAN, LSGAN, WGAN) and robust regularizations (none, GP, AR, RFM). Best viewed on screen.

et al. (2017) showed that the FID metric corresponds well with human judgment of image quality. In our experiments, we use FID as the primary quantitative measure of image quality. As we are mainly interested in the relative improvements of different methods we did not extensively tune the architecture. See the supplementary for additional details on experimental setup.

Table 1 shows the quantitative results. Without any regularization the original and least squares objective work best, while the linear (WGAN) objective does not train well. This is not surprising, as an unconstrained linear objective easily large discriminator outputs and unstable gradients. Adding instance noise (IN) exacerbates this problem for CIFAR-10, but slightly helps for Celeb-A. The gradient penalty (GP) fixes some of the instabilities for both CIFAR-10 and Celeb-A, and helps most for the linear (WGAN) model. However, the gradient penalty does not nearly perform as well as our robust regularizations (AR and RFM), both of which improve the quality of the synthesized images throughout different GAN objectives. Figures 3 and 4 show the corresponding qualitative results. More generated images as well as additional results measuring the effect of different adversarial attacks can be found in supplementary material.

| loss | regularization | | | | | regularization | | | | |
|---|---|---|---|---|---|---|---|---|---|---|
| | None | IN | GP | AR | RFM | None | IN | GP | AR | RFM |
| GAN | 34.25 | 32.28 | 30.22 | **24.06** | 24.50 | 34.63 | 14.23 | 19.61 | **12.04** | 12.15 |
| LSGAN | 32.03 | 32.89 | 29.72 | 27.42 | **25.22** | 36.27 | 14.06 | 21.56 | 15.60 | **11.45** |
| WGAN | 41.58 | 49.80 | 26.20 | 28.42 | **24.90** | 53.50 | 19.41 | 16.12 | 14.16 | **13.40** |
| | (a) FID on CIFAR10 | | | | | (b) FID on Celeb-A | | | | |

Table 1: FID scores on CIFAR10 and FID on Celeb-A for various GAN losses and regularizations. Lower is better for FID.

**Stability across architectures/hyperparameters** We follow Gulrajani et al. (2017) and evaluate the robustness of our regularization across a variety of experimental setups on CIFAR-10. We randomly sample 55 different experiment setups from a large pool of commonly used losses (JS,

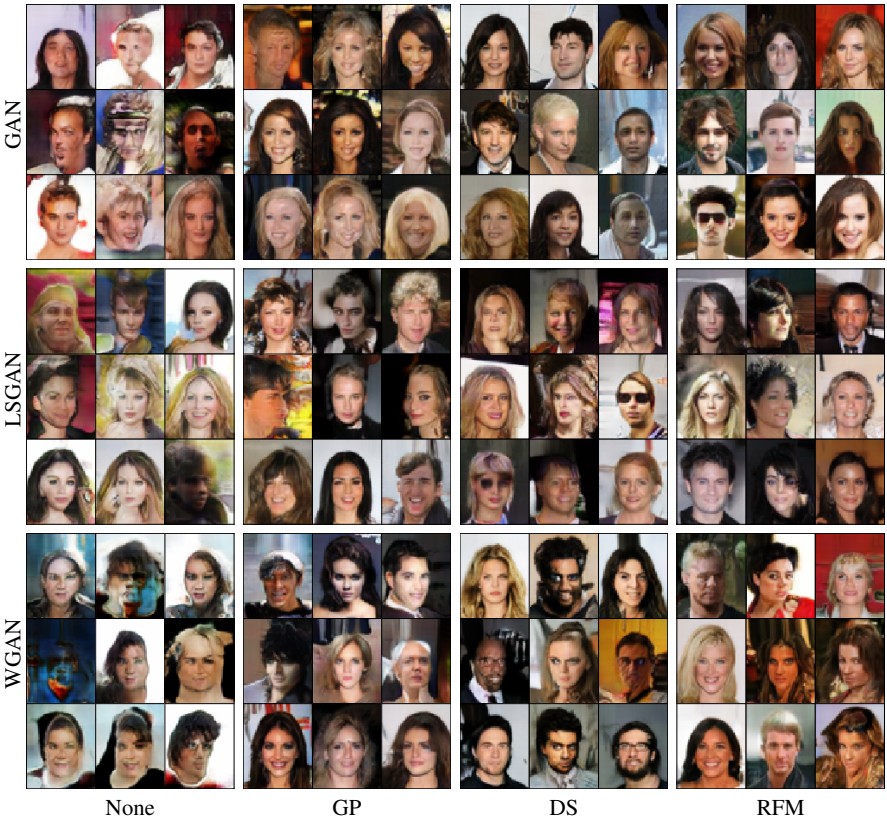

Figure 4: Random image samples from CelebA for various GAN objectives (GAN, LSGAN, WGAN) and robust regularizations (none, GP, AR, RFM). Best viewed on screen.

LSGAN, WGAN), batch size (8, 64), learning rate ($10^{-3}$, $10^{-4}$), network blocks (vanilla convolution, ResNet), filter size (32, 128), nonlinearities (ReLU, LeakyReLU, Tanh) and normalizations (BatchNorm, InstanceNorm, none). For each random setup, we train four separate models: without regularization, with additive instance noise, with gradient penalty, and with RFM. We rank them from 1 to 4 in order of increasing FID score. We then count how many times a given method performs the best and worst out of the batch. Table 2 shows the results. Our robust feature matching consistently outperform other regularization, performing best in over two thirds of the experimental setups.

Experimentally, a robust loss or penalty leads to better convergence, see supplement. However, the theoretical implications of adversarial robustness on local convergence is not yet well understood and warrants further investigation.

|  | None | Noise | GP | RFM |
|---|---|---|---|---|
| # Top Perf. | 1 | 6 | 11 | 37 |
| # Worst Perf. | 39 | 9 | 4 | 3 |
| Average Rank | 3.6 | 2.7 | 2.2 | 1.5 |

Table 2: Performance under different experimental setups.

## 6 CONCLUSION

In this paper, we established a clear connection between robust discriminators in generative adversarial networks and the overall smoothness of the optimization and the quality of the results. To our knowledge, we are the first to show that a robustness regularization guarantees a smooth and robust loss function for any GAN objective. Finally, our results suggest that robust regularization leads to better training and visual results than standard gradient penalties.

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

## APPENDIX A  CONTINUITY AND LIPSCHITZNESS

We start by showing that any GAN with a continuous generator and disciminator has itself a continuous objective.

**Corollary A.1.** If generator, discriminator and loss function $f_F$ in a GAN are continuous, then the GAN objective $\ell(G(\theta))$ is continuous in the generator parameters $\theta$.

*Proof.* By definition of continuity we have $\|G(z; \theta) - G(z; \theta + \Delta_\theta)\| < \varepsilon_G$ for all $\|\Delta_\theta\| < \delta_\theta$ and all noise samples $z$, and $|f_F(D(x)) - f_F(D(x + \Delta_x))| < \varepsilon_D$ for all $\|\Delta_x\| < \delta_x$. Using Theorem 4.1, we know $|\ell(G(\theta)) - \ell(G(\theta) + u)| < \varepsilon_D$ for any perturbation $u(z) = G(z; \theta + \Delta_\theta) - G(z; \theta)$ as long as $\|u(z)\| < \delta_x$. Hence, $|\ell(G(\theta)) - \ell(G(\theta + \Delta_\theta))| < \varepsilon_D$ for all $\|\Delta_\theta\| < \delta_\theta$, if $\delta_x = \varepsilon_G$.  $\square$

The same properties hold for Lipschitz functions.

**Corollary A.2.** In a GAN, if the generator is K-Lipschitz and the discriminator is L-Lipschitz, then the GAN objective is (KLC)-Lipschitz in the generator parameters $\theta$, where $C$ is the expected slope of the loss function.

*Proof.* By definition of Lipschitz continuity any perturbation $u(z) = G(z; \theta + \Delta_\theta) - G(z; \theta)$ is bounded $\|u(z)\| < K\|\Delta_\theta\|$. Furthermore the composition of Lipschitz functions is bounded by $|D(G(z; \theta + \Delta_\theta)) - D(G(z; \theta))| < KL\|\Delta_x\|$. Combining $\Delta_x = u(z)$ with Theorem 4.2 yields $|\ell(G(\theta)) - \ell(G(\theta + \Delta_\theta))| < KLC\|\Delta_\theta\|$. Hence the GAN objective is (KLC)-Lipschitz.  $\square$

## APPENDIX B  ROBUST FEATURE MATCHING BOUND

Here we show that for a optimal discriminator $D^*(x) = w^\top \phi(x)$ in robust feature matching

$$\rho_r(D, G) = E_{z \sim P_z} \left[ \max_{v: \|v\|_p < \delta} \|\phi(G(z)) - \phi(G(z) + v)\|_2^2 \right] + \alpha \|w\|_2^2 \tag{6}$$

is robust.

**Theorem B.1.** For a non-positive loss $f$ and a robust feature matching objective

$$\underset{D \in \mathcal{D}}{\text{maximize}} \, L(D, G) - \lambda \rho_r(D, G)$$

the optimal discriminator $D^*$ is robust

$$E_{x \sim P_Z} \left[ |D^*(G(z)) - D^*(G(z) + u(z))| \right] \leq \frac{-f(0)}{\sqrt{\alpha \lambda}}$$

for all $\|u(z)\|_p < \delta$.

*Proof.* We know that the objective value of the optimal discriminator is larger than the objective of a constant zero discriminator:

$$L(D^*, G) - \lambda \rho_r(D^*, G) \geq L(0, G) - \lambda \rho_r(0, G) = L(0, G)$$
$$\lambda \rho_r(D^*, G) \leq L(D^*, G) - L(0, G) \leq -L(0, G) = -2f(0),$$

where the last inequality holds due to the non-positively of the objective. For every element of the RFM objective

$$\|\phi(G(z)) - \phi(G(z) + v)\|_2^2 + \alpha \|w\|_2^2 \geq 2\sqrt{\alpha}\|w\|_2 \|\phi(G(z)) - \phi(G(z) + v)\|_2 \quad \text{Square expansion}$$
$$\geq 2\sqrt{\alpha}\|w^\top \phi(G(z)) - w^\top \phi(G(z) + v)\|_2 \quad \text{Cauchy–Schwarz}$$
$$= 2\sqrt{\alpha}|D(G(z)) - D(G(z) + v)|.$$

Hence

$$E_{x \sim P_Z} \left[ |D^*(G(z)) - D^*(G(z) + u(z))| \right] \leq \frac{1}{2\sqrt{\alpha}} \rho_r(D^*, G) \leq \frac{-f(0)}{\sqrt{\alpha \lambda}}.$$

$\square$

## APPENDIX C   NETWORK ARCHITECTURE

We use a network architecture almost identical to WGAN-GP for consistency. The generator employs pre-activation batch normalization with ReLU, while the discriminator uses instance normalization with Leaky ReLU for better gradients. We have run experiments using a variety of architectures, including DCGAN, which WGAN-GP builds off of. We find that the results are similar and decided on the architecture from WGAN-GP for the computation efficiency due to the constant filter size.

| Layer | Output shape |
|---|---|
| Latent vector | 128 |
| Linear | 2048 |
| Reshape | $4 \times 4 \times 128$ |
| Residual Block | $4 \times 4 \times 128$ |
| Residual Block | $4 \times 4 \times 128$ |
| Upsample | $8 \times 8 \times 128$ |
| Residual Block | $8 \times 8 \times 128$ |
| Upsample | $16 \times 16 \times 128$ |
| Residual Block | $16 \times 16 \times 128$ |
| Upsample | $32 \times 32 \times 128$ |
| Residual Block | $32 \times 32 \times 128$ |
| Conv 1x1 | $32 \times 32 \times 3$ |
| Sigmoid | $32 \times 32 \times 3$ |

(a) Generator architecture.

| Layer | Output shape |
|---|---|
| Image | $32 \times 32 \times 3$ |
| Conv 3x3 | $32 \times 32 \times 128$ |
| Residual Block | $32 \times 32 \times 128$ |
| Downsample | $16 \times 16 \times 128$ |
| Residual Block | $16 \times 16 \times 128$ |
| Downsample | $8 \times 8 \times 128$ |
| Residual Block | $8 \times 8 \times 128$ |
| Downsample | $4 \times 4 \times 128$ |
| Residual Block | $4 \times 4 \times 128$ |
| Residual Block | $4 \times 4 \times 128$ |
| Residual Block | $4 \times 4 \times 128$ |
| Residual Block | $4 \times 4 \times 128$ |
| Global Average Pooling | 128 |
| Linear | 1 |

(b) Discriminator architecture.

Table 3: GAN Architecture for our experiments on CIFAR-10 and CelebA. Note that CelebA follows the same architecture, but the generator has a few more upsampling and processing blocks to output an image of the correct size.

## APPENDIX D   USING A ROBUST LOSS

Here, we present additional quantitative and qualitative results for robust loss $f$, instead of discriminator. For all regularization, we penalize the gradient or robustness of the loss directly, instead of the discriminator output. The results here agree with our theoretical analysis that a robust loss is beneficial for the generator. However, here the gradient penalty performs worse for GAN and LSGAN, which might be due to vanishing gradients in the loss function close to an optimal discriminator.

| loss | regularization | | |
|---|---|---|---|
| | None | GP | AR |
| GAN | 34.25 | 50.66 | **24.06** |
| LSGAN | 32.03 | 46.99 | **28.33** |
| WGAN | 41.58 | **26.20** | 28.42 |

(a) FID on CIFAR10

| | regularization | | |
|---|---|---|---|
| | None | GP | AR |
| | 34.63 | 53.25 | **11.96** |
| | 36.27 | 57.01 | **15.42** |
| | 53.50 | 16.12 | **14.74** |

(b) FID on Celeb-A

Table 4: FID scores using a robust loss.

## APPENDIX E    DIRAC-GAN

To better understand the effects that a robust loss or discriminator have on the GAN objective function, we use the Dirac-GAN setup. Each example in Figure 5 shows the vector field of the given GAN objective as a function of $(\theta, \psi)$, where the initial points $\theta_0 = \psi_0 = 1$, and the global optimum $\theta^* = \psi^* = 0$. We ran simultanious updates, for an unconstrained GAN, a GAN with hard robustness constraints, and with a robustness regularization. The unregularized GAN does not converge. The hard constraint GAN comes within distance $d$ of the global optimum, where $d$ depends on the strength of the robustness. The smaller $\varepsilon$ the smaller is $d$. Finally, adversarial regularization leads to a convergent Dirac-GAN objective. However, a thorough theoretical analysis is warranted to fully understand this local convergence property.

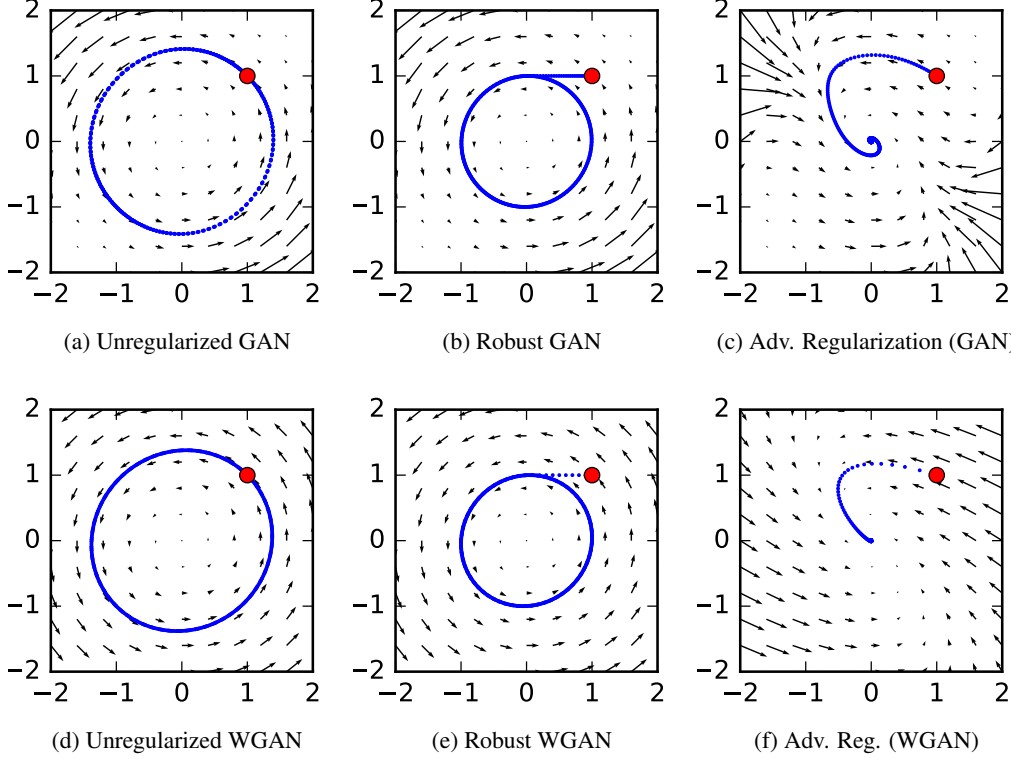

Figure 5: Convergence of Dirac-GAN using various regularization methods (or none) optimized with alternating gradient descent.

APPENDIX F    EFFECTS OF USING DEFENSES AGAINST DIFFERENT ATTACKS

In our main experiments we use an FGM attack (scaled normalized gradient) for simplicity, as other attacks have more hyperparameters such as number of iterations and constants. To show, that this is a result not exclusive to FGM, we provide additional experiments using defenses against other popular attacks - Fast Gradient Sign Method (FGSM), Projected Gradient Descent (PGD) and Carlini Wagner (CW) using robust feature matching (RFM) and adversarial regularization (AR). As expected, these defense techniques also provide improvements to the FID score.

| | FGSM +AR | FGSM +RFM | PGD +AR | PGD +RFM | CW +AR | CW +RFM |
|---|---|---|---|---|---|---|
| GAN (JS) | **23.94** | 26.92 | 26.00 | 27.43 | 24.72 | 30.35 |
| LSGAN | 27.26 | **25.34** | 28.88 | 26.59 | 25.43 | 28.02 |
| WGAN | 29.80 | **25.22** | 30.16 | 29.32 | 40.57 | 27.90 |

Figure 6: FID for additional defenses on CIFAR-10. Results are compatible with Table 2(a) in submission.

## APPENDIX G    ADDITIONAL GENERATED SAMPLES

Finally, we present additional generations for various models.

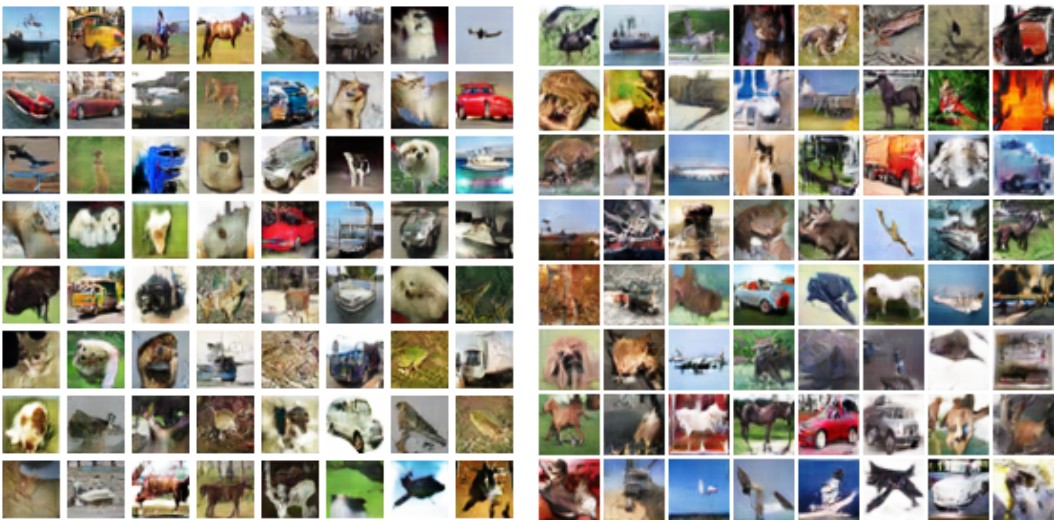

(a) Robust discriminator using WGAN+RFM.                    (b) Robust loss using GAN+DS.

Figure 7: Comparison of the best models from training using a robust discriminator and robust loss.

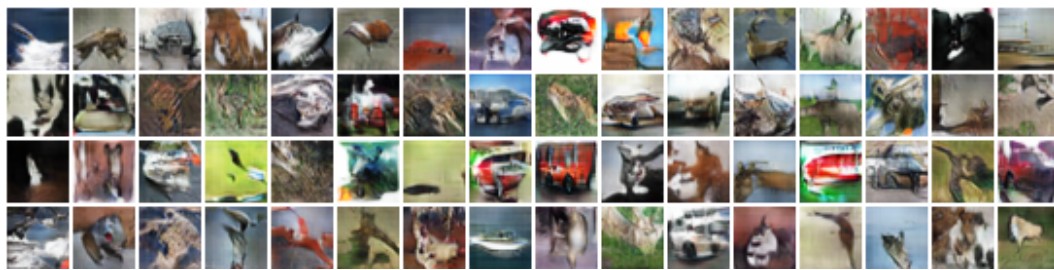

Figure 8: Original GAN.

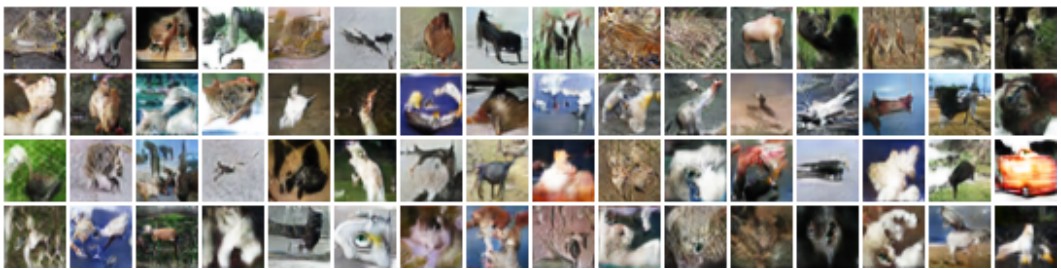

Figure 9: Original LSGAN.

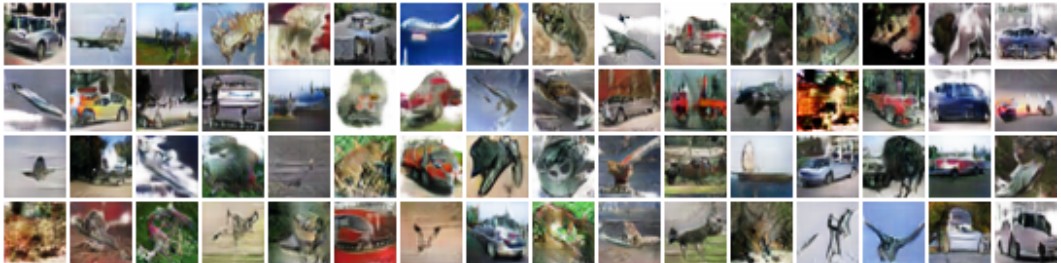

Figure 10: Original WGAN-GP.

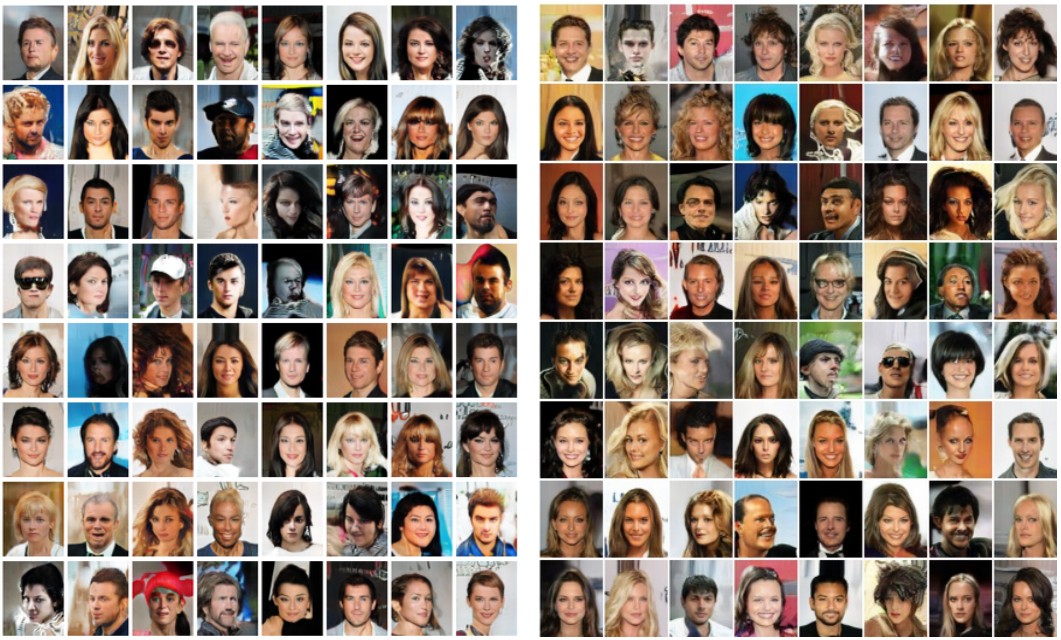

(a) Robust discriminator using WGAN+RFM.  (b) Robust loss using GAN+RFM.

Figure 11: Comparison of the best models from training using a robust discriminator and robust loss.

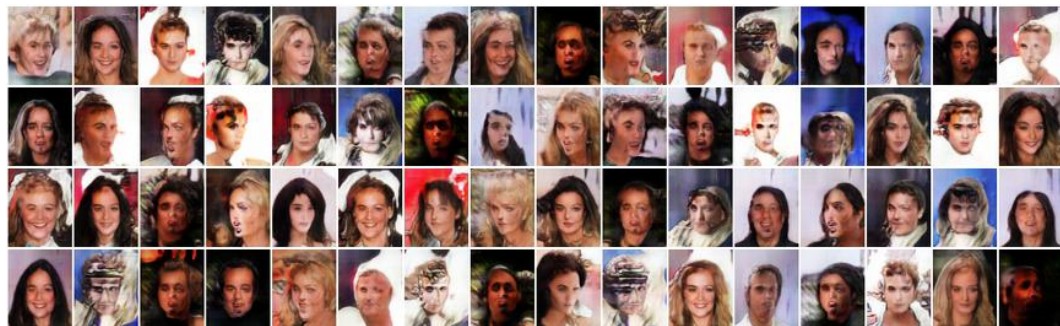

Figure 12: Original GAN.

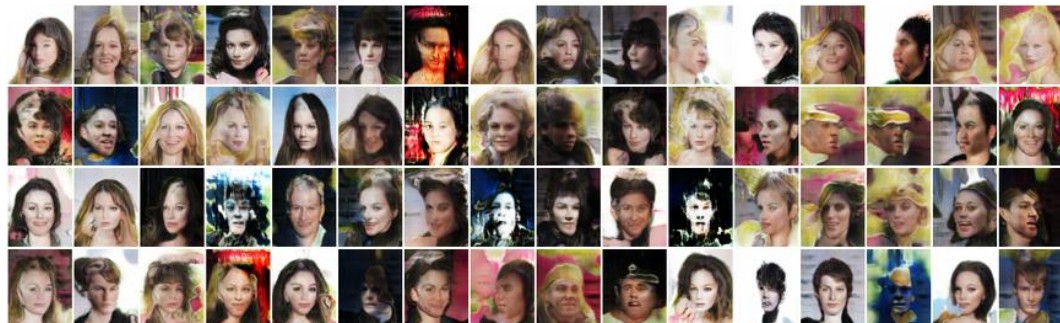

Figure 13: Original LSGAN.

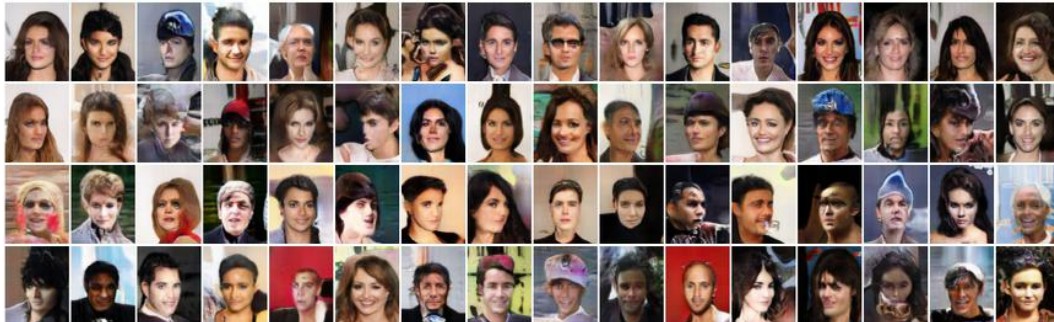

Figure 14: Original WGAN-GP.

