# OpenReview forum: "Don't let your Discriminator  be fooled"
_ICLR.cc/2019/Conference_

### Official Review · AnonReviewer1 · 2018-11-02
**A novel idea, but lack of motivation and intuition**

**Rating:** 6
**Confidence:** 4

**Review:**

## Overview

This paper proposes a new way to stabilize the training process of GAN by regularizing the Discriminator to be robust to adversarial examples. Specifically, this paper proves that a discriminator which is robust to adversarial attacks also leads to a robust minimax objects. Authors provide theoretical analysis about the how the robustness of the Discriminator affects the properties of the objective function, and the proposed regularization term provides an efficient and effective way to regularize the discriminator to be robust. However, it does not build connection between the robustness of the Discriminator and why it can provide meaningful gradient to the Generator. Experimental results demonstrate the effectiveness of the proposed method. This paper is easy to understand.


## Drawbacks
There are some problems in this paper. First, this paper is not highly motivated and lacks of intuition. I can hardly understand why the robustness can stabilize the training of GAN. Will it solve the problem of gradient vanishing problem or speed up the convergence of GAN? The toy example in Sec. 4.2 shows that it can regularize the Discriminator to provide a meaningful gradient to Generator, but no theoretical analysis is provided. The main gap between them is that the smoothness of D around the generated data points does not imply the effectiveness of gradients. Second, the theoretical analysis is inconsistent with the experimental settings. Theorem 4.3 holds true when f is non-positive, but WGAN’s loss function can be positive and this paper does not give any details about this part. Third, in Sec. 4.2, I can hardly distinguish the difference between robust loss, robust discriminator and regularized objectives.

Besides, there are lots of typos in this paper. In Sec 3, Generative Adversarial Networks part, the notations of x and z are quiet confusing. In Definition 3.2, d which measures the distance between network outputs is not appeared above.

## Summarization
Generally, this paper provides a novel way to stabilize the training of GAN. However, it does not illustrate its motivation clearly and no insight is provided.

## After rebuttal
Some of the issues are addressed. So I change my rating to 6.

---

> ### Author Response · Authors · 2018-11-08
> **RE: A novel idea, but lack of motivation and intuition**
>
> We thank the reviewer for her/his time and the constructive feedback. We are glad that the reviewer sees our contribution as a novel idea. We address the main concerns below.
>
> > this paper is not highly motivated and lacks (of) intuition. I can hardly understand why the robustness can stabilize the training of GAN.
>
> As the reviewer rightly pointed out the connection between smoothness of the objective and ease (or stability) of training is an empirical one.
> “The toy example in Sec. 4.2 shows that it can regularize the Discriminator to provide a meaningful gradient to Generator, but no theoretical analysis is provided”
> We are not the first paper to establish this empirical connection, WGAN and follow-up work already established this on a wide range of generative tasks. However, we are the first to point out that this smoothness / robustness is not a property of WGAN, but rather the regularization used to optimize the discriminator in any GAN.
>
> In addition, we show empirically in Table 2 that robustness leads to much more stable training (and better generation performance) than theoretically motivated stability results such as Instance Noise. However, the reviewer is right that there is no theoretical connection, which is an important avenue of future work, but beyond the scope of this paper.
>
>
> > the theoretical analysis is inconsistent with the experimental settings. Theorem 4.3 holds true when f is non-positive, but WGAN’s loss function can be positive and this paper does not give any details about this part.
>
> The reviewer is right that Theorem 4.3 only applies to the JS (GAN) and LS (LSGAN) objectives for which the regularization works best. Theorem 4.3 does not say anything about linear (WGAN) objectives. For WGAN style objectives the original WGAN paper showed robustness results under slightly different conditions. We tried to extend our results to linear objectives, but did not yet succeed (we could not proof or disproof Theorem 4.3 for linear objectives).
> We still included WGAN results in the results to establish the empirical connection between regularization and robustness. While we can not prove robustness for linear objectives, it still holds in practice.
> However, if the reviewer finds this distracting and confusing we are happy to edit or remove parts of the experimental section to make it more consistent.
>
> > in Sec. 4.2, I can hardly distinguish the difference between robust loss, robust discriminator and regularized objectives.
>
> Both robust loss and discriminator pose a hard constraint on the discriminator (either before or after the loss function). These hard constraints are difficult to optimize (see WGAN), but easy to analyze. The regularized objective is easy to optimize as a regularization (soft constraint) between two generative distributions (original and perturbed). Theorem 4.3 shows that the regularized objective can be reduced to hard constraints for the JS and LS objectives, and thus benefits for all the analysis of the hard constraints.
>
> We will update the paper to better highlight this difference.
>
> > Typos and notation
> We thank the reviewer for pointing the typos and notational inconsistencies out, and will fix them in the next iteration.

---

> > ### Comment · AnonReviewer1 · 2018-11-27
> > **Changing rating to 6**
> >
> > Thanks for the detailed feedback. Some of my issues are addressed in the feedback and it would be better to clarify them in the revised paper. Now I change my rating to 6. The reason why I cannot give 7 is the missing analysis of robust D leading to better G.

---

### Official Review · AnonReviewer2 · 2018-11-02
**Good paper with good results**

**Rating:** 7
**Confidence:** 3

**Review:**

The main idea that this paper presents is that making a discriminator robust to adversarial perturbations the GAN objective can be made smooth which results in better results both visually and in terms of FID. In addition to the  proposed adversarial regularisation the authors also propose a much stronger regularisation called robust feature matching which uses the features of the second last layer of the discriminator. I find the ideas presented in this paper interesting and novel.
The authors' claims are supported with sufficient theory and several experiments that prove their claims. The presented results show consistent improvements in terms of FID and actually some of the improvements reported are impressive

---

> ### Author Response · Authors · 2018-11-08
> **RE: Good paper with good results**
>
> We thank the reviewer for the feedback. We are glad you liked the paper.

---

### Official Review · AnonReviewer3 · 2018-11-02
**robustness regularization improves GANs training**

**Rating:** 7
**Confidence:** 3

**Review:**

The paper proposed a systematic way of training GANs with robustness regularization terms. Using the proposed method, training GANs is smoother and

pros
- The paper is solving an important problem of training GANs in a robust manner. The idea of designing regularization terms is also explored in other domains of computer vision research, and it's nice to see the its power in training GANs.
- The paper provides detailed proofs and analysis of the approach, and visualizations of the regularization term help people to understand the ideas.
- The presentation of the approach makes sense, and experimental results using several different GANs methods and competing regularization methods are extensive and good in general

cons
- I didn't find major issues of the paper. I think code in the paper should be made public as it could potentially be very useful for training GANs in general.

---

> ### Author Response · Authors · 2018-11-08
> **RE: robustness regularization improves GANs training**
>
> We thank the reviewer for the insightful feedback. We’re glad the reviewer liked the paper. We will make the code public upon acceptance.

---

### Meta-Review · Area_Chair1 · 2018-12-14
**Good practical approach to stabilise GAN training**

**Confidence:** 4
**Recommendation:** Accept (Poster)

**Metareview:**

The paper provides a simple method for regularising and robustifying GAN training. Always appreciated contribution to GANs. :-)